# A Physics-preserved Transfer Learning Method for Differential Equations

## Abstract

While data-driven methods such as neural operator have achieved great success in solving differential equations (DEs), they suffer from domain shift problems caused by different learning environments (with data bias or equation changes), which can be alleviated by transfer learning (TL). However, existing TL methods adopted in DEs problems lack either generalizability in general DEs problems or physics preservation during training. In this work, we focus on a general transfer learning method that adaptively correct the domain shift and preserve physical information. Mathematically, we characterize the data domain as product distribution and the essential problems as distribution bias and operator bias. A Physics-preserved Optimal Tensor Transport (POTT) method that simultaneously admits generalizability to common DEs and physics preservation of specific problem is proposed to adapt the data-driven model to target domain utilizing the pushforward distribution induced by the POTT map. Extensive experiments demonstrate the superior performance, generalizability and physics preservation of the proposed POTT method.

## 1. Introduction

Many scientific problems, such as climate forecasting (Wu et al., 2023; Verma et al., 2024) and industrial design (Zhou et al., 2024; Borrel-Jensen et al., 2024), are modelized by differential equations (DEs). In practice, DEs problems are usually discretized and solved by numerical methods since analytic solutions are hard to obtain for most DEs. However, traditional numerical solvers typically struggle with expensive computation cost and poor generalization ability. Recently, dealing DEs with deep neural network has attracted extensive attention. These methods can be roughly divided into two categories: physics-driven and data-driven. Physics-driven methods such as Physics-Informed Neural Networks (PINNs) (Raissi et al., 2019; Meng et al., 2024) optimize neural networks with objective constructed by equations formulation to approximate the solution function. Although PINNs methods have great interpretability, they also suffer from the poor generalization capability and the requirement of exact formulation of DEs. Data-driven methods such as DeepONet (Lu et al., 2021) and Fourier Neural Operator (FNO) (Li et al., 2021) take the alterable function in the equations as input data and solution function as output data. The generalization capability of neural operator models are markedly improved because they can cope with a family of equations rather than one.

However, data-driven methods are highly dependent on identical assumption of training and testing environments. If the testing data comes from different distribution, model performance may degrade significantly. In practice, however, applying model to a different data distribution is a common requirement, e.g. from simulation data to experiment data (Liu et al., 2023). While it is hard to collect sufficient data from a new data domain, transfer learning (TL) that aims to transfer model from source domain with plenty of data to target domain with inadequate data, is widely adopted in real-world applications (Zhang et al., 2023).

In this work, we carefully analyze the transfer learning settings in DEs problems and modelize the essential problem as distribution bias and operator bias. Given such perspective, we fully investigate the existing TL methods used in DEs problems. Technically, they can be summarized as three types. **(1) Analytic methods** (Desai et al., 2022) induce an analytic expression for parameters of target model, so the model can be adapted to target domain with few samples. Nevertheless, they are only feasible in few problems with nice properties and hence not a general methodology. **(2) Finetuning** (Subramanian et al., 2023) the well-trained source model by target data. It is widely-used in DEs problems with domain shift due to its simplicity. **(3) Domain Adaptation (DA)** (Wang et al., 2024) methods developed in other areas, e.g., computer vision (CV). They typically align feature distributions of source and target domains and remove the domain-specific information, so the aligned feature is domain-invariant and the predictor trained on source feature distribution can generalize to target features. Among

[1]Anonymous Institution, Anonymous City, Anonymous Region, Anonymous Country. Correspondence to: Anonymous Author <anon.email@domain.com>.

Preliminary work. Under review by the International Conference on Machine Learning (ICML). Do not distribute.

Table 1. Benchmarks used for experiments. The 1st column describes the basic information of the equation and the input and output functions of the DEs problem. Figures in $\mathcal{D}_1$, $\mathcal{D}_2$ and $\mathcal{D}_3$ are examples of (input,output) pairs from different domains in the transfer tasks. For 1-d curve plot, the filled regions represent the areas between the curve and the x-coordinate. For 2-d surface plot, the pixel value at each image pixel corresponds to the function value at the sampling point. Brighter color (yellow) indicates larger value.

| EQUATIONS | $\mathcal{D}_1$ | $\mathcal{D}_2$ | $\mathcal{D}_3$ |
|---|---|---|---|
| BURGERS' EQUATION $u_t + uu_x = \nu u_{xx},$ $x \in (0,1), t \in (0,1],$ $\mathcal{G}_\theta : u(x,0) \to u(x,1)$ |  |  |  |
| ADVECTION EQUATION $u_t + \nu u_x = 0,$ $x \in [0,1], t \in (0,1],$ $\mathcal{G}_\theta : u(x,0) \to u(x,t)$ |  |  |  |
| DARCY FLOW $\nabla \cdot (k(x)\nabla u(x)) = 1,$ $x \in [0,1] \times [0,1],$ $\mathcal{G}_\theta : k(x) \to u(x)$ |  |  |  |

these methods, analytic methods and finetuning directly correct the operator bias while the DA methods correct the distribution bias. Nevertheless, when the amount of available target data is limited, directly correcting operator bias by finetuning is insufficient. On the other hand, if we only correct the distribution bias through feature alignment, the physical relation of DEs may not necessarily be valid in the aligned feature space, which will eventually fail the operator bias correction. Therefore, it is important to develop **a general physics-preserved method** that can transfer the model with only a small amount of target data.

Our idea is to characterize the target domain with physics preservation and then fully correct the operator bias. Specifically, we propose the Physics-preserved Optimal Tensor Transport (POTT) method to learn a physics-preserved optimal transport map between source and target domains. Then the target domain together with its physical structure are characterized by the pushforward distribution, which enables a more comprehensive training for model transfer learning. Thus, the model's generalization performance on target domain can be largely improved even when only a small number of target samples are available for training.

We conduct evaluation and analysis experiments on different types of equations with transfer tasks of varying difficulties. Experimental results demonstrate that POTT outperforms existing transfer learning methods used in DEs problems. Our contributions are summarized as follows:

- A detailed analysis of transfer learning for DEs problems is presented, based on which we propose a feasible transfer learning paradigm that simultaneously admits generalizability to general DEs problems and physics preservation of specific problems.

- We propose POTT method to adapt the data-driven model to target domain utilizing the pushforward distribution induced by the POTT map. A dual optimization problem is formulated to explicitly solve the optimal map. The consistency property between the solution and the ideal optimal map is presented.

- POTT shows superior performance on different types of equations with transfer tasks of varying difficulties. Intuitive visualization analysis further supports our discussion on the physics preservation of POTT.

## 2. Related Works

### 2.1. Data-driven methods for DEs problems

In DEs problems, data-driven methods aim to learn the operator between functions. The most widely-used models are DeepONet, FNO and their variants. Lu et al. (2021) proposed DeepONet based on the universal approximation theorem (Chen & Chen, 1995). MIONet (Jin et al., 2022) extends DeepONet to solve problems with multiple input functions and Geom-DeepONet (He et al., 2024) enables DeepONet to deal with parameterized 3D geometries. Li et al. (2021) proposed FNO by approximating integration in the Fourier domain. Geo-FNO (Li et al., 2023a) extends FNO to arbitrary geometries by domain deformations. F-FNO (Tran et al., 2023) enhances FNO by employing factorization in the Fourier domain. Recently, transformer have also been used to construct neural operators (Li et al.,

2023b; Hao et al., 2023; Li et al., 2023c; Wu et al., 2024). Although having achieved great success, these data-driven methods induce a common issue: they are highly dependent on identical assumption of training and testing environments. If the testing environment differs, the performance of the neural operators will significantly degrade.

## 2.2. Transfer learning

Most transfer learning methods are proposed for Unsupervised Domain Adaptation (UDA) with classification task. They align the feature distributions of source and target domain by distribution discrepancy measurement (Long et al., 2015), domain adversarial learning (Ganin et al., 2016; Chen et al., 2022), etc. Recent methods (Chen et al., 2021; Nejjar et al., 2023; Yang et al., 2025) further extend DA to regression settings with continuous variables. For DEs problems, Desai et al. (2022) presents analytic transfer methods for specific equations; finetuning (Xu et al., 2023; Subramanian et al., 2023) and DA methods (Goswami et al., 2022; Wang et al., 2024) are applied in various tasks. However, there isn't a general physics-preserved transfer learning method developed for DEs problems.

## 2.3. Optimal transport

OT has been quite popular in machine learning area (Cuturi, 2013; Courty et al., 2016). To deal with data of large scale and continuous distribution, neural OT (NOT) methods are proposed (Seguy et al., 2018; Korotin et al., 2023). They typically train the neural network to approximate the OT map via constructing objective function by various OT problems (Fan et al., 2023; Asadulaev et al., 2024).

# 3. Analysis and Motivation

## 3.1. Problem Formulation

Now we formally modelize the transfer learning settings for DEs problems. Consider two function space $\mathcal{D}_k, \mathcal{D}_u$ with elements $k : \Omega_k \to \mathbb{R}$, $u : \Omega_u \to \mathbb{R}$. Denote the product spaces as $\mathcal{D} = \mathcal{D}_k \times \mathcal{D}_u$ and $\Omega = \Omega_k \times \Omega_u$. Suppose there exist physical relations within the product space $\mathcal{D}$, which can be characterized as following two forms:

$$\mathcal{F}(k, u) = 0 \text{ (Equation form)} \tag{1}$$
$$\mathcal{G}(k) = u \text{ (Operator form)} \tag{2}$$

Obviously, relation Eq. (2) is the explicit form of the implicit operator mapping determined by Eq. (1), whose existence is theoretically guaranteed under certain conditions. Once the operator $\mathcal{G} : \mathcal{D}_k \to \mathcal{D}_u$ is solved, we can predict the desired physical quantities $u$ for a group of $k$. However, directly solving the implicit function from Eq. (1) is often extremely difficult. In such cases, fitting $\mathcal{G}$ by neural network with collected data set $\{(k, u)\}$ provides a prac-

tical way for numerical approximation, which is exactly the goal of data-driven methods. Here we slightly abuse the notations $k$ and $u$ to represent both functions and their discretized value vectors.

An essential limitation is that the learning of the operator network $\hat{\mathcal{G}}$ depends heavily on the distribution of collected data. Let $P^s, P^t \in \mathcal{P}_\mathcal{D}$ be two product distributions supported on the source and target domain $\mathcal{D}^s, \mathcal{D}^t \subset \mathcal{D}$, respectively. Then the operator trained from them, denoted as $\hat{\mathcal{G}}^s$ and $\hat{\mathcal{G}}^t$, are in fact the approximations of $\mathcal{G}^s := \mathcal{G}|_{\mathcal{D}^s}$ and $\mathcal{G}^t := \mathcal{G}|_{\mathcal{D}^t}$. When distribution shift occurs between $P^s$ and $P^t$, the operator relation also differs. So the transfer learning problem for DEs can be modelized as

$$
\begin{aligned}
P^s(k, u) &\neq P^t(k, u), \quad \text{(Distribution bias)} \\
\implies \quad \mathcal{G}^s &\neq \mathcal{G}^t. \quad\quad\quad \text{(Operator bias)}
\end{aligned}
\tag{3}
$$

In this situations, the model performance usually degrades if $\hat{\mathcal{G}}^s$ is applied on $\mathcal{D}^t$ directly. While collecting sufficient training data is rather difficult in many applications scenarios, a common requirement is to transfer $\mathcal{G}^s$ to $\mathcal{D}^t$ with a small amount of target domain data available.

Formally, given $\hat{\mathcal{D}}^s = \{(k_i^s, u_i^s)\}_{i=1}^{n^s}$, $\hat{\mathcal{D}}^t = \{(k_j^t, u_j^t)\}_{j=1}^{n^t}$, where $n^t \ll n^s$, the task is to transfer source model $\hat{\mathcal{G}}^s$ to target domain $\mathcal{D}^t$ with collected $\hat{\mathcal{D}}^t$ and approximate the physical relation $\mathcal{G}^t$, i.e. to correct the operator bias.

## 3.2. Methodology Analysis

Based on problem 3, existing transfer learning methods either directly correct the operator bias or indirectly correct the operator bias by aligning the feature distributions, all of which are subject to certain limitations.

**Analytic methods** directly correct the operator bias by deriving analytic expressions

$$\hat{\mathcal{G}}^t = \mathcal{H}_a(\hat{\mathcal{G}}^s, \hat{\mathcal{D}}^t), \tag{4}$$

where $\mathcal{H}_a$ denotes the ideal analytic formulation. Although such methods exhibit excellent interpretability, they require precise physical priors such as the explicit form of the equation, which is extremely difficult for most of DEs problems. Therefore, they can only be applied to a limited amount of problems with well-behaved equations.

**Finetuning by target domain data** also directly corrects the operator bias, which only further trains the source model with collected target data:

$$\hat{\mathcal{G}}^t = \min_{\mathcal{G}} \mathcal{L}_{\text{task}}(\hat{\mathcal{D}}^t; \hat{\mathcal{G}}^s), \tag{5}$$

where $\mathcal{L}_{\text{task}}$ denotes the task-specific training loss. It does not actively and fully leverage the knowledge of available source and target domain data. More importantly, when the

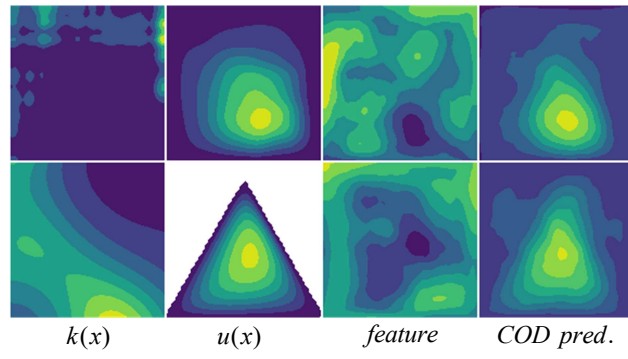

$$k(x) \qquad u(x) \qquad feature \qquad COD\ pred.$$

*Figure 1.* Visualization of COD in task $\mathcal{D}_3 \to \mathcal{D}_2$ on Darcy flow. Figures in the 1st row and 2nd row come from the source domain $\mathcal{D}_3$ and the target domain $\mathcal{D}_2$ respectively. The 1st and 2nd columns present the input and output function sample pairs. The 3rd column shows the visualization of the corresponding feature maps from the aligned feature distributions of samples in 1st column. Features from source and target are analogous but confused. And the physical relation may not necessarily be valid in the aligned feature space. The prediction shown in the 4th column shows that the physical structures of $u$ are not fully preserved.

amount of available target data is limited, the predictions of target samples often exhibit characteristics similar to those of the source samples because small amount of data is not enough for parameters of the model to adapt to the target domain, as discussed in Sec. 5.4.

**Distribution alignment methods from DA** indirectly correct the operator bias by aligning the feature distributions. These methods typically aim to learn a feature map and a corresponding feature space in which the source and target feature distributions are aligned. Therefore, a predictor trained by the source domain features can be expected to perform well on target domain features:

$$g^* = \min_{g}\ \mathrm{dist}(g_{\#}P^s, g_{\#}P^t)$$
$$\hat{\mathcal{G}}^t = \min_{\mathcal{G}}\ \mathcal{L}_{\mathrm{task}}(g^*(\hat{\mathcal{D}}^s) \cup g^*(\hat{\mathcal{D}}^t); \hat{\mathcal{G}}^s), \qquad (6)$$

where $\mathrm{dist}(\cdot, \cdot)$ denotes a measurement of distribution discrepancy, g is the learned feature map, $g_{\#}P^s, g_{\#}P^t$ are the pushforward feature distributions. DA method is purely data-driven without the need of physical priors so it is a general methodology that can be used in most scenarios. However, the two feature distributions are aligned by removing the domain specific knowledge so the domain invariant representations are obtained. In other words, the aligned feature distribution may lose some domain specific physical relations of both domain. As shown in Fig. 3.2, the features of source and target samples are analogous but confused. It is unclear whether the physical relations are preserved, which is also indicated by the output of learned $\hat{\mathcal{G}}^t$.

**Motivation of POTT.** Generally speaking, apart from analytic methods that are not visable in most cases, directly

correcting operator bias by finetuning only partially transfers with limited target data, while indirectly operator bias via feature distribution alignment may distort the physical relations of the DEs problem. Therefore, we propose to correct the operator bias by characterizing the target domain with physics preservation and then fully adapting $\hat{\mathcal{G}}^t$ to $\mathcal{D}^t$:

$$\hat{\mathcal{D}}^r = \mathcal{H}_c(\hat{\mathcal{D}}^s, \hat{\mathcal{D}}^t, \mathcal{R})$$
$$\hat{\mathcal{G}}^t = \min_{\mathcal{G}}\ \mathcal{L}_{\mathrm{task}}(\hat{\mathcal{D}}^r \cup \hat{\mathcal{D}}^t; \hat{\mathcal{G}}^s), \qquad (7)$$

where $\mathcal{H}_c$ is the formulation that characterizes $\mathcal{D}^t$ , $\hat{\mathcal{D}}^r$ is an approximation of $\mathcal{D}^t$ and $\mathcal{R}$ is the physical regularization.

## 4. POTT Method

The major obstacle in Eq. (7) is to construct the $\mathcal{H}_c$. A practical way is to learn a physics-preserved map $\mathcal{T}$ between the product distributions $P^s(k, u)$ and $P^t(k, u)$, so the target distribution $P^t$ can be characterized by the pushforward distribution $P^r = \mathcal{T}_{\#}P^s$, i.e. $\mathcal{H}_c(\cdot) = \mathcal{T}(\hat{\mathcal{D}}^s)$.

While the exact corresponding relations between samples from $P^s$ and $P^t$ is unknown, it is impractical to train $\mathcal{T}$ by traditional supervised learning. In other words, $\mathcal{T}$ shall be trained via an unpaired sample transformation paradigm.

In optimal transport (OT) theory, an optimal transport map between two distributions is the solution of a OT problem and the computation of the OT problem does not need paired samples. Therefore, a natural idea is to modelize the ideal map $\mathcal{T}$ by an optimal transport map between $P^s$ and $P^t$ with physical regularization. In the following sections we regard the desired map as the OT map $\mathcal{T}$, distinguishing from map $T$ that is not necessary optimal.

### 4.1. Formulation of POTT

The most widely known OT problems are the Monge problem and the Kantorovich problem defined as follows:

$$M(P^s, P^t) = \inf_{T_{\#}P^s = P^t} \int_{\Omega^s} c(x, T(x))\, dP^s(x) \qquad (8)$$

$$K(P^s, P^t) = \inf_{\pi \in \Pi(P^s, P^t)} \int_{\Omega^s \times \Omega^t} c(x, y)\, d\pi(x, y), \quad (9)$$

where $c(x, y)$ denotes the cost of transporting $x \in \Omega^s$ to $y \in \Omega^t$, $T : \Omega^s \to \Omega^t$ denotes the transport map, $T_{\#}P^s$ denotes the pushforward distribution, and $\Pi(P^s, P^t)$ denotes the set of joint distributions with marginals $P^s$ and $P^t$.

The Monge problem aims at a transport map $\mathcal{T}$ that minimize the total transport cost, which is called the Monge map. However, usually the solution of the Monge problem does not exist, so the relaxed Kantorovich poblem is more widely used. The Kantorovich problem can be solved as a linear programming problem. With an entropic regularization added, it can be fastly computed via

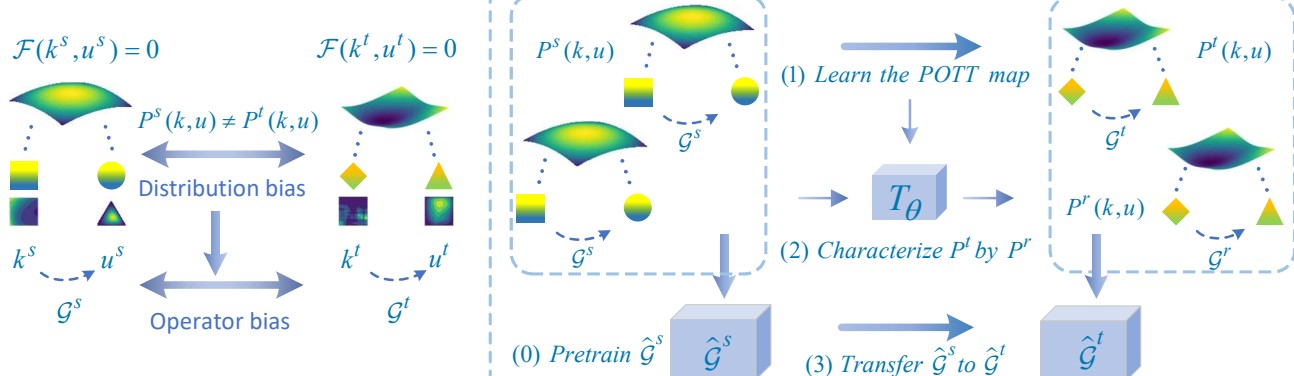

*Figure 2.* Illustration of POTT. **Left:** Illustration of problem formulation. The distribution bias leads to the bias of physics, i.e. the operator bias. The task of transfer learning is to correct the operator bias. **Right:** POTT correct the operator bias by characterizing $\mathcal{D}^t$ in a physics-preserved way. (0) Before the model transfer process, source model $\hat{\mathcal{G}}^s$ is pretrained with sufficient source data. (1) The POTT map $T_\theta$ between source and target domain is firstly learned. (2) The target distribution $P^t$ is characterized by pushforward distribution $P^r$. (3) $\hat{\mathcal{G}}^s$ is adapted to $\hat{\mathcal{G}}^t$ with $\hat{\mathcal{D}}^t$ and $\hat{\mathcal{D}}^r$.

the Sinkhorn algorithm. Moreover, if $\pi^*$ takes the form $\pi^* = [id, \mathcal{T}]_\# P^s \in \Pi(P^s, P^t)$, then $\mathcal{T}$ is the Monge map.

Our purpose is to characterize $P^t$ by $P^r = T_\# P^s$, so we focus on the Monge problem. Actually, the physical relation in DEs problems is a constraint between the marginal distributions $P_k$ and $P_u$. Thus, rather than consider a common Monge problem, a more reasonable perspective is to consider an OT problem between two product distributions, which is known as the Optimal Tensor Transport (OTT) problem. Under this perspective, we propose the physics-preserved optimal tensor transport (POTT) problem:

**Definition 4.1** (POTT). Given two product distributions $P^s, P^t \in \mathcal{P}_{\mathcal{D}_k \times \mathcal{D}_u}$, define the physics-preserved optimal tensor transport (POTT) problem as:

$$\inf_{T_\# P^s = P^t} \int_{\mathcal{D}^s} c\left((k,u), T(k,u)\right) dP^s + \mathcal{R}(T_k, T_u), \quad (10)$$

where $T = (T_k, T_u)$, $T_k = T|_{\mathcal{D}_k}$, $T_u = T|_{\mathcal{D}_u}$, the physical regularization $\mathcal{R}(T_k, T_u)$ depends on the physical relation between the marginal distribution of $P^r$.

Specifically, when the pushforward distribution $P^r$ perfectly matches the target distribution $P^t$, the physical relation within is also obtained. However, it is hard to achieve in practice and $P^r$ should be regarded as an approximation or a disturbance of $P^t$. Then we expect $P^r$ to preserve the physical relation or, in other words, to approach $P^t$ in a physics-preserved way. We proposed a form of the physical regularization as

$$\mathcal{R}(T_k, T_u) = m(\mathcal{G}(k^r), u^r)$$
$$= m(\mathcal{G}T_k(k^s), T_u(u^s)), \quad (11)$$

where the $m(\cdot, \cdot)$ is a metric on $\mathcal{D}_u$, e.g. the $L2$ norm.

### 4.2. Optimization and Analysis

Existing OTT methods (Kerdoncuff et al., 2022) mainly focus on the discrete case of entropy-regularized OTT and solve the optimization problem via the Sinkhorn algorithm, which is not suitable for OTT with physical regularization. Motivated by NOT methods, we explicitly fit $\mathcal{T}$ by a neural network. A common way to optimize a neural network is to update the parameters by the gradient of training loss. But Eq. (10) is a constrained optimization problem and it is challenging to satisfy the constraint during the optimization process. Therefore, we introduce the Lagrange multiplier and reformulate Eq. (10) to the unconstrained dual form.

$$\sup_f \inf_T \int_{\Omega^s} c\left((k,u), T(k,u)\right) - f(T(k,u))$$
$$+ \lambda m(\mathcal{G}T_k(k^s), T_u(u^s)) dP^s + \int_{\Omega^t} f(k,u) dP^t. \quad (12)$$

Optimization with gradient descent tends to converge to a saddle point $(T^*, f^*)$. Following previous work (Fan et al., 2023), we can verify the consistency between $T^*$ and $\mathcal{T}$.

**Theorem 4.2** (Consistency). *Suppose the dual problem Eq. (12) admits at least one saddle point solution, denoted as $(T^*, f^*)$. Let $\mathcal{L}$ be the objective of Eq. (12). Then*

- *the dual problem Eq. (12) equals to the Kantorovich problem with physical regularization in terms of total cost, i.e. $\mathcal{L}(P^s, T^*, f^*) = K(P^s, P^t) + \mathcal{R}(T^*)$.*

- *if $T^*_\# P^s = P^t$, then Eq. (12) degenerates to the dual form of the primal Monge problem Eq. (8), $T^*$ is a Monge map, i.e. $\mathcal{L}(P^s, T^*, f^*) = M(P^s, P^t)$.*

The proof of Thm 4.2 can be found in Appendix B. Theoretically, if the Monge map exists, i.e. $P^r = P^t$, then $P^r$

*Table 2.* Evaluation results of Burgers' equations. The source and target domain of each task are shown in the 1st row. The amount of available target data is shown in the 2nd row. Relative MSE is recorded. The smaller value is better. The best result of each task is in bold.

| | $\mathcal{D}_1 \rightarrow \mathcal{D}_2$ | | $\mathcal{D}_1 \rightarrow \mathcal{D}_3$ | | $\mathcal{D}_3 \rightarrow \mathcal{D}_2$ | | AVERAGE | |
| METHOD | 50 | 100 | 50 | 100 | 50 | 100 | 50 | 100 |
|---|---|---|---|---|---|---|---|---|
| TARGET ONLY | 0.1842 | 0.1229 | 0.1173 | 0.0968 | 0.1842 | 0.1229 | 0.1619 | 0.1142 |
| SOURCE+TARGET | 0.3960 | 0.3982 | 0.3486 | 0.3350 | 0.3145 | 0.2930 | 0.3530 | 0.3421 |
| FINETUNING | 0.2001 | 0.1191 | 0.1049 | 0.0801 | 0.1546 | 0.0938 | 0.1532 | 0.0977 |
| TL-DEEPONET | 0.1623 | 0.1182 | 0.1275 | 0.1127 | 0.1763 | 0.1436 | 0.1554 | 0.1248 |
| DARE-GRAM | 0.1727 | 0.1145 | 0.1241 | 0.1099 | 0.1752 | 0.1393 | 0.1573 | 0.1212 |
| COD | 0.1713 | 0.1225 | 0.1288 | 0.1105 | 0.1818 | 0.1525 | 0.1606 | 0.1285 |
| POTM | **0.1528** | **0.0965** | **0.0950** | **0.0705** | **0.1249** | **0.0757** | **0.1242** | **0.0809** |

automatically admits the physics contained in $P^t$. The solution of the Monge problem is the desired physics-preserved OT map. However, as mentioned in Sec. 4.1, in most cases the Monge problem has no solution and the Monge map does not exists. Therefore, the saddle point $T^*$ is not an optimal solution of the physics-regularized Monge problem. In this situation, Thm 4.2 states that Eq. (12) equals to physics-regularized Kantorovich problem in terms of total cost. Thus, the learned $T^*$ can be regarded as a compromise solution between the Monge problem and the Kantorovich problem. Importantly, the physical regularization encourages physics preservation during the training process of $T$, which is crucial for the approximation of $P^t$. Further illustration can be found in Sec. 5.5.

In practice, we parametrize the map $T$, dual multiplier $f$ and the operator $\mathcal{G}$ by neural networks $T_\theta$, $f_\phi$ and $\mathcal{G}_\eta$ with parameters denoted by $\theta$, $\phi$ and $\eta$. $k$ and $u$ are discretized into vectors and an alternative of the metric $m(\cdot, \cdot)$ is the $L2$ norm in the vector space. Besides, the operator $\mathcal{G}$ in physical regularization term Eq. (11) is substituted by the finetuned $\hat{\mathcal{G}}_\eta^t$ as an approximation. So the overall objective of POTT method is

$$\max_\phi \min_\theta \sum_{i=1}^{n^s} c((k_i^s, u_i^s), T_\theta(k_i^s, u_i^s)) - f_\phi(T_\theta(k_i^s, u_i^s))$$

$$+ \lambda \|\hat{\mathcal{G}}_\eta^t k_i^r - u_i^r\|_2^2 + \sum_{j=1}^{n^t} f_\phi(k_j^t, u_j^t)$$

$$\min_\eta \sum_{j=1}^{n^t} \hat{\mathcal{L}}_{\text{task}}(\hat{\mathcal{G}}_\eta^t(k_j^t), u_j^t) + \beta \sum_{i=1}^{n^s} \hat{\mathcal{L}}_{task}(\hat{\mathcal{G}}_\eta^t(k_i^r), u_i^r),$$

$$(13)$$

where $k_i^r = T_{\theta_k}(k_i^s), u_i^r = T_{\theta_u}(u_i^s)$. $\hat{\mathcal{L}}_{\text{task}}$ is the task specific loss. $\lambda$ and $\beta$ are hyper-parameters. An intuitive illustration is shown in Fig. 4.1.

# 5. Experiment

We conduct evaluation and analysis experiments on three different equations with transfer tasks of varying difficulties. Implementation details are provided in Appendix C.

## 5.1. Benchmarks

Following previous works, we take the 1-d Burgers' equation, 1-d space-time Advection equation, and 2-d Darcy Flow problem as our benchmarks. A brief introduction of these DEs problems and the transfer tasks can be found in Tab. 1. To simulate scenarios of domain shift, we adjusted the sampling distribution of input functions and the parameters of the equations to generate three different domains for each equation, denoted as $\mathcal{D}_1$, $\mathcal{D}_2$ and $\mathcal{D}_3$. We generate 1000 training samples for each domain of Burgers' equation and 2000 samples for Advection equation and Darcy Flow. Model trained with such number of samples can be regarded as oracle of the domain. To fully investigate the effectiveness of transfer learning methods in DEs problems, we consider two scenarios that only 50 and 100 target data samples are available for training. For all transfer tasks, we use 10 extra target domain samples for validation and 100 for testing. The relative Mean Square Error

$$rMSE = \frac{\|u_{pred} - u_{gt}\|_2^2}{\|u_{gt}\|_2^2} \quad (14)$$

is reported, where $u_{gt}$ denotes the ground truth of output u.

## 5.2. Comparision methods

As discussed in Sec. 3.2, analytic methods are hard to apply in general data-driven DEs methods. So we compare POTT with finetuning and DA methods.

**Finetuning.** Finetuning is the most widely used transfer method in DEs problems. So we regard it as the baseline for comparision. Besides, training from scratch with target data and training from scratch with mixed data from both domains are also included as baseline methods, denoted by **Target only** and **Source+Target**, respectively.

**DA methods.** Both the input and output functions of DEs problems are continuous variables, which means the transfer learning problem should be categorized as Domain Adaptation Regression (DAR) problem. TL-DeepONet (Goswami et al., 2022) is a representative method proposed for DEs problems. Besides, we adopt state-of-the-art DAR methods

Table 3. Evaluation results of Advection equations.

| METHOD | $\mathcal{D}_1 \to \mathcal{D}_2$ | | $\mathcal{D}_2 \to \mathcal{D}_1$ | | $\mathcal{D}_3 \to \mathcal{D}_2$ | | AVERAGE | |
|---|---|---|---|---|---|---|---|---|
| | 50 | 100 | 50 | 100 | 50 | 100 | 50 | 100 |
| TARGET ONLY | 0.2162 | 0.1261 | 0.1585 | 0.1182 | 0.2162 | 0.1261 | 0.1970 | 0.1235 |
| SOURCE+TARGET | 0.2347 | 0.1400 | 0.7299 | 0.4041 | 0.3969 | 0.1574 | 0.4538 | 0.2338 |
| FINETUNING | 0.0247 | 0.0143 | 0.2193 | 0.0891 | 0.1257 | 0.0723 | 0.1532 | 0.0977 |
| TL-DEEPONET | 0.0587 | 0.0127 | 0.2365 | 0.1047 | 0.1534 | 0.0685 | 0.1495 | 0.0620 |
| DARE-GRAM | 0.0572 | 0.0121 | 0.2227 | 0.0805 | 0.1687 | 0.0700 | 0.1495 | 0.0542 |
| COD | 0.0530 | 0.0120 | 0.2252 | **0.0785** | 0.1593 | 0.0644 | 0.1458 | 0.0516 |
| POTT | **0.0207** | **0.0112** | **0.1872** | 0.0787 | **0.1016** | **0.0613** | **0.1032** | **0.0504** |

Table 4. Evaluation results of Darcy flow.

| TASK
METHOD | $\mathcal{D}_2 \to \mathcal{D}_1$ | | $\mathcal{D}_1 \to \mathcal{D}_3$ | | $\mathcal{D}_2 \to \mathcal{D}_3$ | | ADVERGE | |
|---|---|---|---|---|---|---|---|---|
| | 50 | 100 | 50 | 100 | 50 | 100 | 50 | 100 |
| TARGET ONLY | 0.1615 | 0.1122 | 0.2925 | 0.2493 | 0.2925 | 0.2493 | 0.2488 | 0.2036 |
| SOURCE+TARGET | 0.7113 | 0.7600 | 0.1581 | 0.1409 | 0.3535 | 0.2381 | 0.4076 | 0.3797 |
| FINETUNING | 0.1426 | 0.0869 | 0.1556 | 0.1605 | 0.4693 | 0.3553 | 0.2558 | 0.2009 |
| TL-DEEPONET | 0.1410 | 0.0805 | 0.1539 | 0.1481 | 0.4514 | 0.2842 | 0.2488 | 0.1709 |
| DARE-GRAM | 0.1395 | 0.0805 | 0.1533 | 0.1441 | 0.4509 | 0.2842 | 0.2479 | 0.1696 |
| COD | 0.1367 | 0.0794 | 0.1527 | 0.1481 | 0.4437 | 0.2836 | 0.2444 | 0.1704 |
| POTT | **0.1362** | **0.0762** | **0.1397** | **0.1404** | **0.3527** | **0.2271** | **0.2095** | **0.1479** |

DARE-GRAM (Nejjar et al., 2023) and COD (Yang et al., 2025) to further investigate the effectiveness of DA methods. Since DARE-GRAM and COD are unsupervised DAR methods proposed for tasks in computer vision, we add the supervised target loss to them for fairness.

### 5.3. Evaluation result

We evaluate these transfer learning method in some hard tasks that need more proactive transfer.

**Burgers' equation.** As shown in Tab. 2, the improvement of POTT compared to finetuning is substantial. When the amount of target data is only 50, POTT reduced the relative error by 23.64% (from 0.2001 to 0.1528) in task $\mathcal{D}_1 \to \mathcal{D}_2$ and by 19.21% (from 0.1546 to 0.1249) in task $\mathcal{D}_3 \to \mathcal{D}_2$. When the amount of target data is 100, although the performance of finetuning greatly improves, POTT still largely reduces the relative error by 18.98% and 19.30% in task $\mathcal{D}_1 \to \mathcal{D}_2$ and $\mathcal{D}_3 \to \mathcal{D}_2$. In the relatively simple task $\mathcal{D}_1 \to \mathcal{D}_3$, although finetuning already achieves satisfactory results, POTT can still reduce the relative error by about 10%, while TL-DeepONet, DARE-GRAM and COD even caused negative transfer and result in larger relative error.

**Advection equation.** As shown in Tab. 3, POTT significantly reduces the relative error in every task. In task $\mathcal{D}_1 \to \mathcal{D}_2$ with 50 target samples, DAR methods fail to correct the operator bias due to limitation of target data. In contrast, POTT remarkably achieves a reduction of 16.19% compared to finetuning. When the amount of target data comes to 100 and the performance of DAR methods is improved, POTT reduce the relative error to lower value. In

average of all tasks, POTT reduce the relative error by about 16% compared to finetuning.

**Darcy flow.** As shown in Tab. 4, with limited target samples, the relative errors of finetuning and training from scratch are large. Model trained from scratch with mixed data is even misled and the prediction error is very large. In task $\mathcal{D}_2 \to \mathcal{D}_3$ where the two domains differ significantly, the performance of finetuning is not satisfactory. Although DAR methods show great improvement compared to finetuning, their enhancement is limited in the extremely challenging task in which only 50 target samples are available. In contrast, the performance of POTT is very impressive. In the task with 50 target samples, the relative error of POTT decreased by nearly 25% (from 0.4693 to 0.3527), achieving a model capability comparable to that of finetuning with 100 target samples. In the task with 100 target samples, the reduction even reaches 36.08% (from 0.3553 to 0.2271), remarkably surpassing all DAR methods. In tasks $\mathcal{D}_2 \to \mathcal{D}_1$ and $\mathcal{D}_1 \to \mathcal{D}_3$ that are simpler than $\mathcal{D}_2 \to \mathcal{D}_3$, POTT still reduce the relative error compared to finetuning and DAR methods in every transfer task.

In summary, POTT's capacity to improve model's performance is related to the difficulty of the transfer task. In the difficult tasks that finetuning is not satisfying, POTT shows great enhancement on cross-domain generalization.

### 5.4. Visualization analysis of $\hat{\mathcal{G}}^t$

Fig. 5.3 illustrates the outputs and error maps for the same target sample predicted by POTT, finetuning, and COD on the Darcy $\mathcal{D}_3 \to \mathcal{D}_2$ task with 100 target samples.

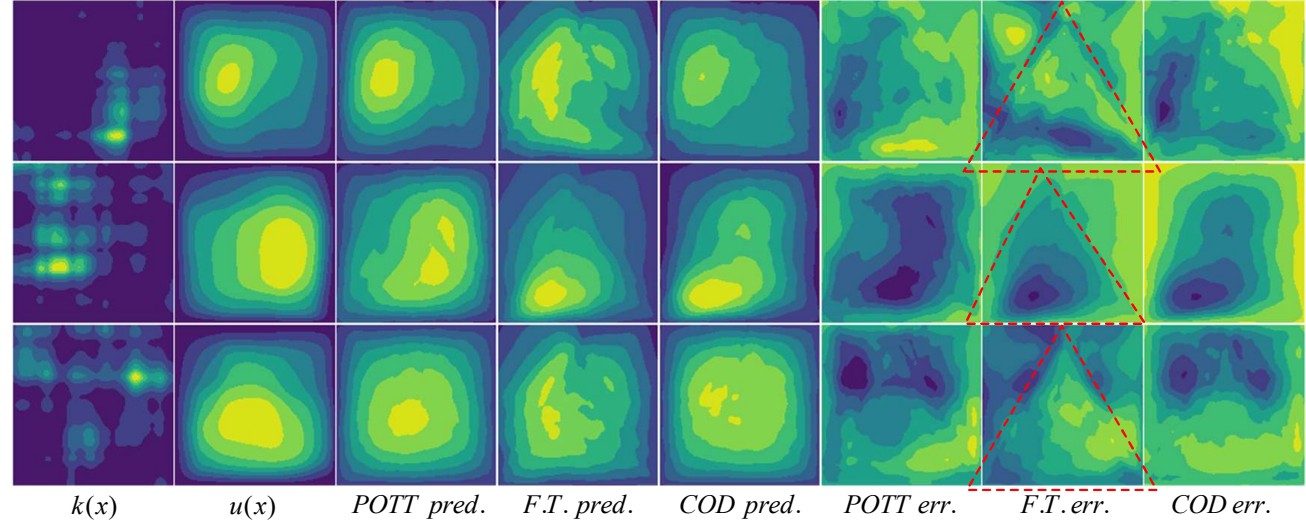

$$k(x) \qquad u(x) \qquad POTT\ pred. \quad F.T.\ pred. \quad COD\ pred. \quad POTT\ err. \qquad F.T.\ err. \qquad COD\ err.$$

*Figure 3.* Visualization of learned Operator network $\hat{\mathcal{G}}_t$ on Darcy Flow. Since the input and output functions in Darcy Flow are two-dimensional, they can be intuitively visualized as images, where the pixel value at each image pixel corresponds to the function value at the sampling point. Brighter colors (yellow) indicate higher values. The 1st and 2nd columns show the input and output function pair from target domain. The 3rd, 4th and 5th columns show the output functions predicted by POTT, finetuning (denoted as F.T. for short) and COD, respectively. The 6th, 7th and 8th columns show the error between the predictions and the ground truth.

As shown in 3rd, 4th and 5th columns, despite only 100 target samples are available for training, outputs predicted by POTT are consistent with the ground truth globally. The shape and the variation trend are quite similar to the ground truth. By contrast, predictions of finetuning indicate that the transferred model fails to learn the right shape and variation trend. Outputs predicted by COD are relatively consistent with ground truth globally. However, the transferred model fails to correctly predict the large value area.

In the error maps shown in 6th, 7th and 8th columns, the bright areas in the error maps of POTT are the smallest among the three methods. Especially, the error maps of finetuning shown in 7th column clearly exhibit the characteristics of source domain distribution, i.e. the distinct triangular patterns, which strongly supports the discussions in Sec. 3.2. Besides, as shown in 8th column, the bright yellow areas in the error maps of COD are larger than POTT. This indicates that although the feature distribution learned by COD has certain similarity to the target domain, it does not fully preserve the physical structure of the function, resulting in the loss of some physical information.

### 5.5. Ablation analysis of physical regularization

To investigate the effect of the physical regularization in Eq. (10), we implement an ablation analysis on task $\mathcal{D}_1 \rightarrow \mathcal{D}_2$ of Darcy flow with 100 target samples. As shown in Fig. 5.4, the outputs of OTT map roughly shape like the ground truth in large value area. However, the triangular structure of target sample is not preserved, indicating the loss of some physics structure. In contrast, with physical regularization, the outputs of POTT map exhibit similarity

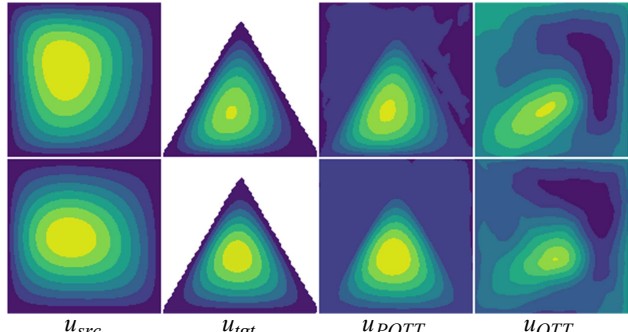

$$u_{src} \qquad u_{tgt} \qquad u_{POTT} \qquad u_{OTT}$$

*Figure 4.* Visualization of POTT map and OTT map on Darcy Flow. Images in 1st and 2nd columns are selected $u(x)$ pairs from source and target domain. The 3rd and 4th columns show the transported $u(x)$ from source samples to target domain via POTT map and OTT map respectively.

in the large value area as well as consistency of the triangular structure, verifying the preservation of physics.

## 6. Conclusion

In this work, we studied the domain shift issue in DEs problems, the essential problems of which are modelized as distribution bias and operator bias. Then we detailedly analyzed existing TL methods adopted in DEs problems and propose a feasible POTT method that simultaneously admits generalizability to common DEs and physics preservation of specific problem. A dual optimization problem together with the consistency property is formulated to explicitly solve the optimal map. Numerical evaluation and analysis validated the effectiveness of POTT.

## Impact Statement

This paper presents work whose goal is to advance the field of Machine Learning. There are many potential societal consequences of our work, none which we feel must be specifically highlighted here.

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

## A. Notations

The notations appear in this paper are summarized as follows:

*Table 5.* Notations.

| Symbols | Meaning |
|---|---|
| $\mathcal{D} = \mathcal{D}_k \times \mathcal{D}_u = \{(k, u)\}$ | General function set |
| $\Omega = \Omega_k \times \Omega_u$ | Domain of function |
| $k = k(x)$ | function defined on $\Omega_k$ |
| $u = u(x)$ | function defined on $\Omega_u$ |
| $\mathcal{F}(x; k, u)$ | Differential Equation |
| $\mathcal{G} : \mathcal{D}_k \to \mathcal{D}_u$ | Operator map from $\mathcal{D}_k$ to $\mathcal{D}_u$ |
| $P(k, u)$ | Product probability function defined on function set $\mathcal{D}$ |
| $p(k, u)$ | Product probability density function of function set $\mathcal{D}$ |
| $\mathcal{P}_{\mathcal{D}}$ | Set of probability functions defined on $\mathcal{D}$ |
| $\mathcal{D}^s$ | Soure domain; a subset of $\mathcal{D}$ |
| $\mathcal{D}^t$ | Target domain; a subset of $\mathcal{D}$ |
| $\mathcal{G}^s$ | Operator relation on $\mathcal{D}^s$ |
| $\mathcal{G}^t$ | Operator relation on $\mathcal{D}^t$ |
| $P^s$ | Probability function on $\mathcal{D}^s$ |
| $P^t$ | Probability function on $\mathcal{D}^t$ |
| $T : \mathcal{D}^s \to \mathcal{D}^t$ | Function map between $\mathcal{D}^s$ and $\mathcal{D}^t$ |
| $\mathcal{T}$ | Ideal solution of optimal transport problem |
| $M(P^s, P^t)$ | Monge problem between probability $P^s$ and $P^t$ |
| $K(P^s, P^t)$ | Kantorovich problem between probability $P^s$ and $P^t$ |
| $\mathcal{R}(T)$ | Physical regularization on $T$ |
| $m(\cdot, \cdot)$ | Metric defined on $\mathcal{D}_u$ |
| $M_{phy}(P^s, P^t)$ | Monge problem with physical regularization |
| $K_{phy}(P^s, P^t)$ | Kantorovich problem with physical regularization |
| $c(\cdot, \cdot)$ | Cost function in OT problem |
| $f$ | Lagrange multiplier |
| $\mathcal{L}$ | Objective function of optimization problem |
| $(T^*, f^*)$ | Saddle point solution of dual problem |
| $\hat{\mathcal{G}}^s$ | Approximated operator on $\mathcal{D}^s$ |
| $\hat{\mathcal{G}}^t$ | Approximated operator on $\mathcal{D}^t$ |
| $\hat{T}$ | Approximation of $T$ |

we slightly abuse the notations $k$ and $u$ to represent both functions and their discretized value vectors. The superscript s or t denotes the domain. The subscript $k$ or $u$ denotes the projection of the original product space or distribution.

## B. Theory

### B.1. Derivation of Eq. (12)

Given POTT problem

$$\inf_{T_\# P^s = P^t} \int_{\mathcal{D}^s} c\left((k, u), T(k, u)\right) dP^s + \mathcal{R}(T_k, T_u), \tag{15}$$

where $T = (T_k, T_u), T_k = T|_{\mathcal{D}_k}, T_u = T|_{\mathcal{D}_u}$, we reorganize it as a constrained optimization problem:

$$\inf_{T} \int_{\mathcal{D}^s} c\left((k, u), T(k, u)\right) dP^s + \mathcal{R}(T_k, T_u) \tag{16}$$

$$s.t. T_\# P^s = P^t \tag{17}$$

Following the dual optimization theory, we introduce the Lagrange multiplier $f$ to construct the Lagrange function:

$$\mathcal{L}(T, f) = \int_{\Omega_k \times \Omega_u} c\left((k, u), T(k, u)\right) dP^s + \mathcal{R}(T_k, T_u) + \int_{\Omega_k \times \Omega_u} f(k, u) d(P^t - T_\# P^s). \tag{18}$$

Substituting the physics regularization by Eq. (11), it comes to

$$\mathcal{L}(T, f) = \int_{\Omega_k \times \Omega_u} c\left((k, u), T(k, u)\right) - f(T(k, u)) + \lambda d(GT_k(k^s), T_u(u^s)) dP^s + \int_{\Omega_k \times \Omega_u} f(k, u) dP^t. \quad (19)$$

Thus, the dual problem of Eq. (15) is

$$\sup_f \inf_T \ \mathcal{L}(T, f), \quad (20)$$

which is exactly Eq. (12).

## B.2. Proof of Thm. 4.2

We prove Thm. 4.2 based on previous work by Fan et al. (2023).

**Theorem B.1** (Consistency). *Suppose the dual problem Eq.* (12) *admits at least one saddle point solution, denoted as* $(T^*, f^*)$. *Let* $\mathcal{L}$ *be the objective of Eq.* (12). *Then*

- *the dual problem Eq.* (12) *equals to the Kantorovich problem with physical regularization in terms of total cost,* *i.e.* $\mathcal{L}(P^s, T^*, f^*) = K(P^s, P^t) + \mathcal{R}(T^*)$.

- *if* $T^*_{\#} P^s = P^t$, *then Eq.* (12) *degenerates to the dual form of the primal Monge problem Eq.* (8), $T^*$ *is a Monge map,* *i.e.* $\mathcal{L}(P^s, T^*, f^*) = M(P^s, P^t)$.

*Proof.* **(1)** Let $k^r = T_{\theta_k}(k^s), u^r = T_{\theta_u}(u^s)$, the inner optimization problem can be formulated as

$$\inf_T \mathcal{L}(T, f) = \inf_T \int_{\Omega_k \times \Omega_u} c\left((k^s, u^s), T(k^s, u^s)\right) - f(T(k^s, u^s)) + \lambda m(Gk^r, u^r) dP^s + \int_{\Omega_k \times \Omega_u} f(k^t, u^t) dP^t$$

$$= -\int_{\Omega_k \times \Omega_u} \sup_{(\xi^r, \zeta^r)} \{ f(\xi^r, \zeta^r) - [c\left((k^s, u^s), (\xi^r, \zeta^r)\right) + \lambda m(G\xi^r, \zeta^r)] \} dP^s + \int_{\Omega_k \times \Omega_u} f(k^t, u^t) dP^t \quad (21)$$

$$= \int_{\Omega_k \times \Omega_u} f(k^t, u^t) dP^t - \int_{\Omega_k \times \Omega_u} f^{c,-}(k^s, u^s) dP^s,$$

where

$$f^{c,-}(k^s, u^s) = \sup_{(\xi^r, \zeta^r)} \left( f(\xi^r, \zeta^r) - [c\left((k^s, u^s), (\xi^r, \zeta^r)\right) + \lambda m(G\xi^r, \zeta^r)] \right) \quad (22)$$

is the c-transform of the physics-regularized Kantorovich dual problem. Then the optimization problem Eq. (12) becomes

$$\sup_f \left[ \int_{\Omega_k \times \Omega_u} f(k^t, u^t) dP^t - \int_{\Omega_k \times \Omega_u} f^{c,-}(k^s, u^s) dP^s \right], \quad (23)$$

which is exactly the physics-regularized Kantorovich problem.

Therefore, if $(T^*, f^*)$ is the saddle point solution of Eq. (12), then $f^*$ is an optimal solution of Eq. (23), $\mathcal{L}(P^s, T^*, f^*) = K(P^s, P^t) + \mathcal{R}(T^*)$, which verifies the first assertion of the theorem.

**(2)** The saddle point $(T^*, f^*)$ satisfy

$$T^*(k^s, u^s) \in argmax_{(\xi^r, \zeta^r)} f^*(\xi^r, \zeta^r) - [c\left((k^s, u^s), (\xi^r, \zeta^r)\right) + \lambda m(G\xi^r, \zeta^r)] \ a.s. \quad (24)$$

$$\implies f^{*c,-}(k^s, u^s) = f^*(T^*(k^s, u^s)) - [c\left((k^s, u^s), T^*(k^s, u^s)\right) + \lambda m(GT^*_k k^s, T^*_u u^s)] \quad (25)$$

where $f^{*c,-}(k^s, u^s) = \sup_{(\xi^r, \zeta^r)} \left( f^*(\xi^r, \zeta^r) - [c\left((k^s, u^s), (\xi^r, \zeta^r)\right) + \lambda m(GT^*_k k^s, T^*_u u^s)] \right)$.

With condition $T^*_{\#} P^s = P^t$, the pushforward distribution $P^r = P^t$, then

$$(T^*_k k^s, T^*_u u^s) = (k^t, u^t)$$
$$\implies m(GT^*_k k^s, T^*_u u^s) = m(Gk^t, u^t) = 0. \quad (26)$$

Thus Eq. (12) degenerates to

$$\sup_{f} \inf_{T} \int_{\Omega_k \times \Omega_u} c((k,u), T(k,u)) - f(T(k,u)) dP^s + \int_{\Omega_k \times \Omega_u} f(k,u) dP^t, \tag{27}$$

which is exactly the dual form of the primal Monge problem. Then we have

$$
\begin{aligned}
&\int_{\Omega} c\left((k^s, u^s), T^*(k^s, u^s)\right) dP^s \\
&= \int_{\Omega} f^*(T^*(k^s, u^s)) dP^s - \int_{\Omega} f^{*c,-}(k^s, u^s) dP^s \\
&= \int_{\Omega} f^*(k^t, u^t) dP^t - \int_{\Omega} f^{*c,-}(k^s, u^s) dP^s \\
&= \int_{\Omega \times \Omega} f^*(k^t, u^t) - f^{*c,-}(k^s, u^s) d\pi \\
&\leqslant \int_{\Omega \times \Omega} c\left((k^s, u^s), (k^t, u^t)\right) d\pi, \ \forall \pi \in \Pi(P^s, P^t)
\end{aligned}
\tag{28}
$$

Take infimum on both sides of the inequation, we obtain

$$
\begin{aligned}
&\inf_{T} \int_{\Omega} c\left((k^s, u^s), T^*(k^s, u^s)\right) dP^s \\
&\leqslant \inf_{\pi} \int_{\Omega \times \Omega} c\left((k^s, u^s), (k^t, u^t)\right) d\pi \\
&\leqslant \int_{\Omega} c\left((k^s, u^s), T(k^s, u^s)\right) dP^s,
\end{aligned}
\tag{29}
$$

where $T$ is any map that satisfies $(Id, T)_{\#} P^s = \pi \in \Pi(P^s, P^t)$. Therefore, the solution of the Monge problem exists and $T^*$ is the Monge map. □

## C. Experiment details

### C.1. Equations, transfer tasks and data generation

#### C.1.1. BURGERS' EQUATION

Considering the 1-D Burgers' equation on unit torus:

$$u_t + u u_x = \nu u_{xx}, \ x \in (0,1), t \in (0,1], \tag{30}$$

we aim to learn the operator mapping the initial condition to the solution funciton at time one, i.e. $\mathcal{G}_\theta : u_0 = u(x,0) \to u(x,1)$. We differ the generation of $u_0$ and parameter $\nu$ to construct different domains:

*Table 6.* Generation of $u_0$ and parameter settings in 1-d Burgers' equation.

| SUB-DOMAIN | DESCRIPTION |
|:---:|:---:|
| $\mathcal{D}_1$ | $u_0 \sim \mathcal{N}(0, 7^2(-\Delta + 7^2\mathcal{I})^{-2}), \nu = 0.01$ |
| $\mathcal{D}_2$ | $u_0 \sim \mathcal{N}(0.2, 49^2(-\Delta + 7^2\mathcal{I})^{-2.5}), \nu = 0.002$ |
| $\mathcal{D}_3$ | $u_0 \sim \mathcal{N}(0.5, 625^2(-\Delta + 25^2\mathcal{I})^{-2.5}), \nu = 0.004$ |

The $\mathcal{N}$ denotes the normal distribution. The resolution of x-axis is 1024.

#### C.1.2. ADVECTION EQUATION

The Advection equation takes the form

$$u_t + \nu u_x = 0, \ x \in (0,1), t \in (0,1]. \tag{31}$$

We aim to learn the operator mapping the initial condition to the solution funciton at a continuous time set $[0, 1]$, i.e. $\mathcal{G}_\theta :$ $u_0 = u(x, 0) \rightarrow u(x, t)$. We differ the function types and generation process of $u_0$ and parameter $\nu$ to construct different domains: The $\mathcal{U}$ denotes the uniform distribution. The resolution of x-axis and t-axis are 100 and 50, respectively.

*Table 7.* Generation of $u_0$ and parameter settings in Advection equation.

| SUB-DOMAIN | DESCRIPTION |
|:---:|:---:|
| $\mathcal{D}_1$ | $u_0(x) = ax^2 + bx + c, \quad a, b, c \in \mathcal{U}(-1, 1), \nu = 3$ |
| $\mathcal{D}_2$ | $u_0(x) = ax^3 + bx^2 + cx + d, \quad a \in \mathcal{U}(0, 1), b, c \in \mathcal{U}(-0.5, 0.5), d = 0.5, \nu = 2$ |
| $\mathcal{D}_3$ | $u_0(x) = a\sin(bx + c), \quad a \in \mathcal{U}(0, 1), b \in \mathcal{U}(5, 10), c \in \mathcal{U}(-1, 1), \nu = 1$ |

### C.1.3. DARCY FLOW

The 2-d Darcy flow takes the form

$$\nabla \cdot (k(x)\nabla u(x)) = 1, \ x \in [0, 1] \times [0, 1]$$
$$u(x) = 0, \ x \in \partial([0, 1] \times [0, 1]). \tag{32}$$

We aim to learn the operator mapping the diffusion coefficient $k(x)$ to the solution function $u(x)$, i.e. $\mathcal{G}_\theta : k(x) \rightarrow u(x)$. We use the leading 100 terms in a truncated $Karhunen - Lo\grave{e}ve$ (KL) expansion for a Gaussian process with zero mean and covariance kernel $\mathcal{K}(x)$ to generate $a(x)$, and construct different function domains by differ the kernel $\mathcal{K}(x, x')$ The

*Table 8.* Generation of $a(x)$ in Darcy flow.

| SUB-DOMAIN | DESCRIPTION |
|:---:|:---:|
| $\mathcal{D}_1$ | $\mathcal{K}(x, x') = exp(-\frac{\|x-x'\|_2^2}{2})$, $\Omega_u$ IS A SQUARE WITH VERTICE ON $\{(0,0), (0,1), (1,0), (1,1)\}$ IN $[0, 1] \times [0, 1]$ |
| $\mathcal{D}_2$ | $\mathcal{K}(x, x') = exp(-\frac{\|x-x'\|_2^2}{2})$, $\Omega_u$ IS A TRIANGLE WITH VERTICE ON $\{(0,0), (0,1), (0.5,1)\}$ IN $[0, 1] \times [0, 1]$ |
| $\mathcal{D}_3$ | $\mathcal{K}(x, x') = exp(-\frac{\|x-x'\|_1^2}{2})$, $\Omega_u$ IS A SQUARE WITH VERTICE ON $\{(0,0), (0,1), (1,0), (1,1)\}$ IN $[0, 1] \times [0, 1]$ |

resolution of $x \in [0, 1] \times [0, 1]$ is $64 \times 64$.

All data are generated based on code provided by Li et al. (2021) and Lu et al. (2022).

### C.2. Implementation details

To test the generalizability of POTT with different models, we employed different backbones on various datasets. On the Burgers' equation dataset, $\mathcal{G}_\eta$ is parametrized as a 1-d Fourier Neural Operator (FNO) model, $T_\theta$ is an operator network composed of two fully connected networks (FCN), and $f_\phi$ is an FCN. On the Advection equation and Darcy flow datasets, $\mathcal{G}_\eta$ adopt a 2-d DeepONet model, $T_\theta$ has a structure similar to $\mathcal{G}_\eta$, and $f_\phi$ is a convolutional neural network (CNN). We use Adam as optimizer and the learning rate is $1e - 3$ for all tasks. The learning rate of the backbone of $\mathcal{G}_\eta$ is ten times smaller than the last two layers, which is a widely-used technique in transfer learning. A cosine annealing strategy is adopted for learning rate of $\mathcal{G}_\eta$.

