# OpenReview forum: "A Physics-preserved Transfer Learning Method for Differential Equations"
_ICML.cc/2025/Conference — Submitted to ICML 2025_

### Official Review · Reviewer_BWbB · 2025-03-09

**Overall Recommendation:** 3

**Summary:**

The paper introduces the problem of domain shifts when learning neural operators, particularly when the model needs to be transformed to predict similar PDEs possibly with different input distributions. To improve upon the existing transfer learning approaches, the paper uses optimal tensor transport methods, with added loss term to incorporate a

**Claims And Evidence:**

I am not convinced by the experimental results. See the Experiments section below.

**Essential References Not Discussed:**

I’m not fully convinced that operator learning are inherently data-driven, since physics-informed losses can be added as done in PINNs [1,2]. Because of that, I would also be interested to see whether these methods could outperform those models as well, where target data and a physics-informed loss are used for training.


[1] Physics-Informed Neural Operator for Learning Partial Differential Equations, https://arxiv.org/pdf/2111.03794

[2] Physics-Informed Deep Neural Operator Networks, https://arxiv.org/pdf/2207.05748

**Experimental Designs Or Analyses:**

Major issue with experimental results -- none of the results report any standard deviation scores. This would make the results more convincing especially that some loss terms are quite close, by showing the statistical significance.

Furthermore, the domain shifts chosen for the results can seem a bit cherry-picked -- since there are only 3 dataset, there would be 6 domain shifts available, which should not be much more experiments that need to be run. This would provide more transparent results, and give better interpretation to the Average scores.

Results regarding training time would also be good since they can demonstrate how computationally effieicnt the methods are.

**Methods And Evaluation Criteria:**

The benchmarks seem to include a good selection of PDEs that would already be standard in the area. The different values of $\nu$ of the PDEs seem to be an important case where the re-training of operators may be necessary, and may be worth highlighting further. However, I think they can seem a bit like toy examples, and therefore could benefit from more realistic data scenarios or even with real experimental data if those can be obtained.

**Other Comments Or Suggestions:**

Since this seems to be targeted as a more in the science domain (as mentioned in the primary area), it can probably be written so that it is more friendly towards people unfamiliar with transfer learning. Particularly, it can be further highlighted which details are the more "standard" techniques from transfer learning, and which are the novel area that arises due to the physics-informed portions, and what they may look like wrt actual PDE equations that the practitioner may want to incorporate (I think these are in the paper however can still be made clearer).

**Other Strengths And Weaknesses:**

Strengths: I do think it does provide an interesting perspective on incorporating domain knowledge into domain adaptation works, which potentially has use cases outside of learning physics-obeying neural operators. So even if not necessarily practical in physics domains, I think it can still be a valuable work nonetheless.

Weaknesses: discussed in the paper, but mainly (1) use in the physics-related domains, and (2) aspects of the experimental results.

**Questions For Authors:**

I haven't really phrased them as questions but these two points are my concerns related to the motivation of the work:

1. Why is transfer learning really necessary, since all examples seem to focus on known PDEs, so in these cases physics informed loss maybe used, and shifting to a new domain may not even be necessary (mentioned this again in the References Not Discussed section).

2. Furthermore, since the tests seem to focus on simulated data (which most likely can be adjusted to fit new conditions for example), it may be difficult to imagine a scenario where data in the target domain cannot already be generated, especially if the PDE is fully known even in the transferred domain. It may be more convincing to show a more concrete example where this occurs, or maybe through some real dataset where it may be hard to find data in one domain than in another.

**Relation To Broader Scientific Literature:**

I believe that improving the efficiency in learning deep operators is of great scientific importance since it allows amortisation of the simulation of data. However, I think the link as to why transfer learning is the way to do so could be made more clear. See Questions section below.

**Theoretical Claims:**

I skimmed through the proofs and see no immediate issues.

---

> ### Author Rebuttal · Authors · 2025-04-01
>
> Thank you very much for taking the time to review our paper and provide professional and constructive reviews. We are encouraged by the positive comments like "a valuable work" of our work. All suggestions will be carefully incorporated into final version.
>
> Q1. Operator learning and problem settings.
>
> (1) Problem explanation and physics-informed loss for operator learning
>
> Perhaps our experimental setup could be misinterpreted as focusing on learning solution functions for known equations. In fact, as highlighted in the end of the introduction in the paper, our method focuses on the transfer learning of neural operator models that learns function-to-function mappings without requiring explicit equations forms. Unlike analytical models like PINNs, the data-driven operator learning is more suitable for real-world problems where equations are approximations of the problems. For instance, several kinds of differential equations can be used to characterize the weather forecasting problem, but none of them is proved to be the true inherent physical relations. In this case, the physics-informed loss is infeasible, while the data-driven property allows operator learning to be more widely applicable in such scenarios.
>
> (2) Necessity of transfer learning.
>
> Despite its advantages, operator learning assumes identical training and testing environments. If testing data comes from a different distribution, (e.g., cross-region weather forecasting), the trained operator model performance may degrade significantly. In such cases, transfer learning is essential for improving model accuracy in cross-domain deployment. Again, as the exact physical relations within problems are usually unknown, the physics-informed losses is infeasible. In contrast, the physical constraints in Eq. (10) is more flexible and practical for broader applications.
>
> (3) Real-world examples.
>
> As mentioned above, this problem setting indeed has broader applications scenarios. In weather forecasting, for instance, differential equations are used to approximate the weather systems, and operator networks are trained on collected data to fit the systems. Generally, more data is available in developed regions while less data is available in less developed regions. As geographical conditions vary across regions, directly deploying models across regions leads to performance degradation. Thus, introducing transfer learning into cross-domain model deployment provides critical value to these areas.
>
> While industrial data would better validate our method, these data is often difficult to obtain. The standard practice in the community is to use data generated from classic equations for simulations. Models performing well on simulated data are then further developed on real-world data. As our work is one of the pioneer works in combining differential equations problems with general transfer learning, our validation on simulated data implies the potential real-world applicability. And we plan to further explore real-world applications in future work.
>
> Q2. Experimental settings and results.
>
> Due to space constraints, here we take the Darcy flow as example for explanation.
>
> (1)	In fact, the results we reported in the paper is the average results of three times repeated experiments. We didn’t report the standard deviation (std) in the paper due to the space limitations of the tables. Now we provide the std of finetuning (FT) and POTT in Darcy flow for example:
>
> Tasks | $D_2 \to D_1$ | $D_1 \to D_3$ | $D_2 \to D_3$
>
> $N_t$ | 50 | 100 | 50 | 100 | 50 | 100
>
> FT | 0.1426$\pm 0.003$ | 0.0869$\pm 0.001$ | 0.1556$\pm 0.012$ | 0.1605$\pm 0.008$ | 0.4693$\pm 0.034$ | 0.3553$\pm 0.026$
>
> POTT | 0.1362$\pm 0.002$ | 0.0762$\pm 0.002$ | 0. 1397$\pm 0.009$ | 0. 1404$\pm 0.006$ | 0.3527$\pm 0.025$ | 0.2271$\pm 0.019$
>
> As can be seen, the std of FT and POTT is not significant, and POTT still outperforms FT with std taken into consideration.
>
> (2)	As for the concerns about sub-domain selection, as stated at the beginning of Section 5.3, "We evaluate these transfer learning methods in some challenging tasks that require more proactive transfer." For the transfer tasks with minor distribution shift, FT is sufficient to perform well and additional transfer learning methods are unnecessary. And that’s why we only conduct experiments in 3 tasks in each datasets, as you pointed out. We run the remaining experiments in Darcy flow for further explanation:
>
> Tasks | $D_1 \to D_2$ | $D_3 \to D_1$ | $D_3 \to D_2$
>
> $N_t$ | 50 | 100 | 50 | 100 | 50 | 100
>
> FT | 0.0515 | 0.0429 | 0.0492 | 0.0469 | 0.0669 | 0.0598 |
>
> POTT | 0.0496 | 0.0409 | 0.0489 | 0.0445 | 0.0649 | 0.0525 |
>
> It’s obvious that the distribution shifts are minor in these tasks, the performance of FT is good enough and additional transfer learning methods are unnecessary. But noted that the POTT method still outperforms FT to some degree.

---

### Official Review · Reviewer_cQRd · 2025-03-12

**Overall Recommendation:** 3

**Summary:**

The paper proposes a transfer learning method for differential equations (DEs) that preserves physics consistency while adapting models to new domains. By decomposing domain shifts into distribution bias and operator bias, the author introduces Physics-preserved Optimal Tensor Transport (POTT), which learns to map the source and target domains under the physical constraints. This method addresses the challenge of poor generalization in traditional data-driven models when applied to out-of-distribution data. The effectiveness of POTT is demonstrated through experiments on both 1D and 2D DEs, showing its ability to improve predictive performance under domain shifts.

**Claims And Evidence:**

The paper claims that POTT can achieve higher performance in domain-transfer tasks compared to the existing fine-tuning based and domain-adaptation methods. This claim is supported by Table 2, which presents numerical results demonstrating that POTT consistently achieves lower error compared to baseline methods. Additionally, the magnitude of the error across different tested scenarios reinforces the author’s assertion that the generalization ability of the proposed method is related to the scenario difficulties.

**Essential References Not Discussed:**

The author discusses the majority of the essential references.

**Experimental Designs Or Analyses:**

The author designs the experiments to compare POTT against six existing methods, including fine-tuning approaches and domain-adaptation techniques. While POTT is used for data-driven models, since it is specifically utilized to ensure physics preservation, some physics-informed models should be utilized as baselines to demonstrate the method's effectiveness, such as [1] and  [2].

[1] Raissi, Maziar, Paris Perdikaris, and George Em Karniadakis. "Physics informed deep learning (part i): Data-driven solutions of nonlinear partial differential equations." arXiv preprint arXiv:1711.10561 (2017).

[2] Li, Zongyi, et al. "Physics-informed neural operator for learning partial differential equations." ACM/JMS Journal of Data Science 1.3 (2024): 1-27.

**Methods And Evaluation Criteria:**

The author evaluates POTT on three DEs in 1D and 2D scenarios. Each DE is generated using three different distributions, and the results are presented for all possible domain shift scenarios. The evaluation metric used is relative Mean Square Error (rMSE), which is a standard measure in numerical experiments for assessing prediction accuracy in DE solving tasks. While the evaluation is limited to synthetic data, it is suggested to incorporate real-world datasets to better assess the robustness of the proposed domain shift method, such as fluid dynamic or climate data.

**Other Comments Or Suggestions:**

It is suggested to include ablation studies on different backbone model architectures for each task to assess the generalizability of POTT. While the paper demonstrates POTT’s effectiveness using a specific choice of backbone models, evaluating its performance across multiple architectures would provide stronger evidence of its robustness and adaptability to various neural PDE solvers.

**Other Strengths And Weaknesses:**

Clarity strength: The paper has a clear flow and is easy to follow.

**Questions For Authors:**

Question 1: How does the performance of fine-tuning and target-only methods compare to POTT as the amount of target data increases? Specifically, does POTT continue to provide advantages when more target data becomes available, or do fine-tuning and target-only methods eventually close the performance gap?

**Relation To Broader Scientific Literature:**

The paper discusses transfer learning and domain adaptation approaches for scientific machine learning models. Some relevant fields are already discussed in the Related Works section, including data-driven approaches in DE solving, transfer learning, and optimal transport.

**Theoretical Claims:**

Section 4 describes the theoretical formulation of POTT as an optimal transport problem and introduces approximations that ensure the learned mapping transfers the input distribution to the target domain while preserving physical properties.

---

> ### Author Rebuttal · Authors · 2025-04-01
>
> Thank you very much for taking the time to review our paper and provide professional and constructive reviews. We are encouraged by the positive comments on clarity strength of our paper. All suggestions will be carefully incorporated into final version.
>
> Q1. Baselines and backbone
>
> (1)	PINNs, PINO and similar model aim are the learning model in differential equations (DEs) problems. In the discussion on transfer learning in DEs problems, they should be regarded as backbone models like DeepONet and FNO rather than baseline of transfer learning method. In fact, the PINO model mentioned by the reviewer is exactly an enhanced model of the FNO.
>
> What’s more, PINNs and PINO are analytical models requiring explicit equation expressions of the problems, which are often impractical in real-world applications. In fact, in most problems the equations are only the approximations and no exact forms of equations are available to train the PINNs or PINO models. For instance, several kinds of differential equations can be used to characterize the weather forecasting problem, but none of them is proved to be the true inherent physical relations. In this case, the physics-informed loss is infeasible, while the data-driven property allows operator learning model like DeepONet and FNO to be more widely applicable in such scenarios. As our work focuses on developing a general method, we didn’t take the analytical models like PINNs and PINO as backbones in our experiments.
>
> (2)	To assess the generalizability of POTT, we applied it on both DeepONet and FNO in different tasks, as mentioned in the appendix C.2. Such experiment settings is not a cherry-picking strategy that choose a specific backbone for the specific tasks but instead demonstrates the generalizability of POTT. However, as suggested by the reviewer, we plan to include additional ablation studies in the final version to further emphasize its generalizability .
>
> Q2. Relationship between performance comparison and the amount of target data
>
> Our experimental design systematically evaluates how target sample amounts influences model accuracy. Here’s a concise summary:
>
> (1)	When very few target samples are available (maybe only less than 5 target samples), the transfer learning is extremely difficult.
>
> (2)	When the amount of target samples are limited but not extreme (as in our experiment settings), POTT outperform  finetune and existing DA methods, and significantly improve the model performance in target domain.
>
> (3)	When the target samples are abundant as in source domain, even the target-only  supervised training is sufficient, then the transfer learning methods are unnecessary.
>
> Q3. Real-world datasets
>
> Thank you for this important suggestion.
>
> (1)	This problem setting indeed has broad applications in various problem scenarios. In weather forecasting, for instance, differential equations are used to approximate the weather systems, and operator networks are trained on collected data to fit these systems. Generally, more data is available in developed regions while less data is available in less developed regions. As geographical conditions vary across regions, directly deploying models across regions can reduce their performance. Thus, introducing transfer learning into cross-domain model deployment provides critical value to these areas.
>
> (2)	While industrial data would better validate operator models, these data is often difficult to obtain. The standard practice in the community is to use data generated from classic equations for simulations. Models performing well on simulated data are then further developed on real-world industrial data. As our work is one of the pioneer works in combining differential equations problems with general transfer learning, our validation on simulated data aligns with conventions and suggests potential real-world applicability of proposed method. And we plan to further explore the applications of POTT in real-world industrial data in future work as suggested.

---

### Official Review · Reviewer_NHBg · 2025-03-13

**Overall Recommendation:** 3

**Summary:**

The authors propose a method to tackle transfer learning in solving differential equations. Current data-driven methods for solving differential equations suffer when training and testing environments differ, or from insufficient data. While transfer learning (TL) has been used previously to adapt models trained on one domain for another domain, the authors identify two types of bias in current TL strategies, which hinders their generalization capability. The authors propose Physics-preserved Optimal Tensor Transport (POTT), which can simultaneously correct the two types of biases in TL, and improve performance even when the target domain dataset is small. Experimental results are provided to show superior performance of the proposed method, and a theoretical analysis is also presented.

**Claims And Evidence:**

The authors claim that their proposed transfer learning method is more general compared to the baselines, and present visualizations in several figures (1,3,4). However, the figures need major rethinking, as currently it is very hard to get the message clearly from these figures. I suggest the authors put arrows/markings inside the figure, so that readers know where to look and what to look for in these figures. A common way to do this is showing ground truth and predictions side by side so that readers clearly see the match/mismatch. The authors can think about doing something similar, otherwise it is hard to see for example in Figure 3, which method is the proposed one, which are the baselines, and why one is better than the other.

**Essential References Not Discussed:**

None

**Experimental Designs Or Analyses:**

The authors present detailed experimental results, I haven’t checked them thoroughly, however from a glance they look extensive.

**Methods And Evaluation Criteria:**

The methodology and problem formulation of the paper at its current state need some major updates. Here are my suggestions:
1. The problem of neural operator learning (first line of the abstract) needs more description as the whole problem formulation depends on this, perhaps discuss how this is different from function learning with examples as differentiation/integration
2. The problem of operator bias is not intuitively/clearly explained at the beginning, while data shift is known to people working outside of transfer learning. Perhaps the authors can provide a more detailed description at the beginning in the introduction (this is provided later)
3. The meaning of physics-preserved needs a bit more detail. What does it mean to characterize a target domain with physics preservation, aren’t physics laws already preserved in all domains?
4. Problem formulation at present is hard to follow in the middle, during the discussion around Equation 3. Perhaps the authors can add an example of what P_s, D_s looks like in real world scenarios.
5. What is meant by pushforward distribution? This is mentioned several times before details are provided.
6. Where is Figure 3.2 as cited in section 3.2?
7. What is meant by features being “confused”?
8. Minor typo: “visable” in Motivation of POTT, perhaps the authors meant “feasible”?

**Other Comments Or Suggestions:**

None

**Other Strengths And Weaknesses:**

None

**Questions For Authors:**

None

**Relation To Broader Scientific Literature:**

Transfer learning is a topic of general interest in the machine learning community. Solving differential equations is a topic of great interest in the physical science community. The authors identify 2 key issues with current transfer learning strategies, and offer a solution in the context of solving differential equations. Therefore the current work has broad implications in both domains.

**Theoretical Claims:**

I did not check the theoretical claims.

---

> ### Author Rebuttal · Authors · 2025-04-01
>
> Thank you very much for taking the time to review our paper and provide professional and constructive reviews. We are encouraged by the positive comments on the broad implications of our work in physical science and domain adaptation community. All suggestions will be carefully incorporated into final version.
>
> Q1. Figures explanation
>
> We sincerely thank the reviewer for this suggestion. In Figure 3, we did arrange the figures in a "ground truth and predictions side by side" format. As described in the caption of Figure 3: Columns 1 and 2 show the input and output function pairs from the target domain. Columns 3, 4, and 5 display the output functions predicted by POTT, finetuning, and COD, respectively. Columns 6, 7, and 8 show the prediction errors relative to the ground truth. We also added borders in Column 7 to emphasize the source domain function properties in finetuning. To further enhance readability and clarity, we will incorporate clearer arrows/markings in the final version as suggested.
>
> Q2. More description on problem formulation
>
> We thank the reviewer for the suggestions and summarize them as a more detailed explanation on problem formulation:
>
> (1) Operator Learning: Unlike function learning (e.g., PINNs), which learns a specific solution function from equation (e.g., u: x → u(x)), operator learning aims to learn a function-to-function mapping (e.g., G : k(x) → u(x)). This is critical for generalization. For example, in Darcy flow, while function learning models cannot adapt to problems with new permeability coefficients k(x), operator learning models can predict u(x) for varying k(x). Notably, when k(x) is the identity mapping, operator learning reduces to function learning.
>
> (2) Operator Bias: Despite its advantages, operator learning assumes identical training and testing environments. If testing data comes from a different distribution (e.g., varying types of k(x)), the intrinsic physics also differs. Then the trained operator model performance may degrade significantly, which is described as the operator bias in Eq. (3).
>
> (3) Examples: As discussed in Sec. 5.1, to simulate distribution and operator bias, we sampled k(x) from three distinct function distributions $P_1$, $P_2$, $P_3$ to form three subdomains $D_1$, $D_2$, $D_3$ and the corresponding transfer tasks for each datasets. As shown in Tab. 1, functions from different subdomains exhibit clear differences due to their distinct distributions. The examples are thus provided in Tab. 1.
>
> Q3. Aren't physics laws already preserved in all domains? More description on physics-preserved.
>
> Yes, the physical laws are already in target domain. But they are hard to learn with limited target data. While the physical laws are determined by the problems and subdomains properties, in cross-domain model transfer, we often have limited target-domain samples, which are insufficient for the model to learn the underlying physical laws of the target domain. POTT addresses this by leveraging both source and target samples to characterize the target distribution in a physics-preserved way, and thus enabling the model to learn the embedded physical laws in target domain.
>
> Q4. Noun explanation
>
> (1) The pushforward distribution is a concept in probability theory, referring to the distribution obtained by transforming another distribution through a mapping. For example, g_\# $P_{s}$ denotes the distribution obtained by applying the transformation g to $P_s$. Formally:
>
> g_\# $P_{s}(A) = P_s(g^{-1}(A)) $
>
> for all Borel set $A$ in the output space of g.
>
> (2) We describe the features in the 3rd column of Figure 1 as "confused" to indicate that the aligned feature space lacks clear structure. As shown in the figure, these feature maps do not convey explicit physical meaning, and it is unclear whether they retain the correct physical information. Thus, we refer to them as "confused."
>
> Q5. Typo
>
> Thank you for your correction. Figure 3.2 should be figure 1. And the "visable" in line 218 should be "feasible" indeed.

---

> > ### Comment · Reviewer_NHBg · 2025-04-04
> >
> > Thank you to the authors for their explanations, these should be included in the manuscript so that readers can easily understand. I have updated my score, I wish you good luck.

---

> > > ### Author Response · Authors · 2025-04-05
> > >
> > > Thank you very much for your reply and your constructive reviews. We will carefully incorporate all your suggestions into the manuscript.

---

### Official Review · Reviewer_sTtY · 2025-03-14

**Overall Recommendation:** 3

**Summary:**

This paper proposes POTT (Physics-preserved Optimal Tensor Transport), a transfer learning method for differential equations. Instead of preserving PDE structures directly, POTT ensures that the operator relationship u=G(k) is maintained during adaptation via optimal transport (OT) with a physics-regularized term. The method is tested on Burgers’, Advection, and Darcy Flow equations, showing improved generalization over fine-tuning and domain adaptation methods.

**Claims And Evidence:**

The transfer learning tasks and methods are also interesting, but the name of the method ‘physics-preserved’ may be a little misleading. It seems as if it preserves the structure of physical laws and equations, but the reality is that it is regularised so that the solution operator u=G(k) pre-trained with the source domain data is preserved.

**Essential References Not Discussed:**

The authors may need to add a discussion of infinite-dimensional OT.

**Experimental Designs Or Analyses:**

Please add a discussion about calculation time.

**Methods And Evaluation Criteria:**

Although you are using OT for finite-dimensional probability distributions, shouldn't you actually be formulating it using OT on an infinite-dimensional function space [R1]?

[R1] Minh, H.Q. Infinite-dimensional distances and divergences between positive definite operators, Gaussian measures, and Gaussian processes. Info. Geo. (2024).

**Other Comments Or Suggestions:**

N/A

**Other Strengths And Weaknesses:**

Strengths
- The introduction of physics-regularized OT for operator transfer is an interesting extension of standard domain adaptation techniques.
- Many ML-based PDE solvers struggle with generalization in the domain shift settings. This paper addresses the significant problem.
- The proposed method is reasonable.

Weaknesses
- The learning procedure is a little difficult to understand. It would be good to write the steps and procedures for learning G^s, T_\theta, and G^t in the form of an algorithm.
- Computational cost and scalability not discussed. OT problem is generally costly.
- The name of the method ‘physics-preserved’ may be a little misleading. The paper claims to preserve "physics," but POTT does not enforce PDE constraints explicitly.

**Questions For Authors:**

- Can you explain the relationship with infinite-dimensional OT?
- Can you explain the learning procedure in the form of an algorithm?
- Are you conducting experiments with settings where the sub-domains overlap to some extent? I thought that the accuracy would drop if the sub-domains were too far apart.
- Can you tell us how the accuracy changes with the number of samples in the target domain?
- I think it would be good if you could mention important applications for this problem setting. For example, it would be effective for experiments on systems with low energy, which are easy to conduct and for which a lot of data can be collected, but for which there are only a few data points for systems with high energy.

**Relation To Broader Scientific Literature:**

The domain shift problem in operator learning is very important and interesting. Operator learning is a promising approach to accelerate physical simulations.

**Theoretical Claims:**

No problem.

---

> ### Author Rebuttal · Authors · 2025-04-01
>
> Thank you very much for taking the time to review our paper and provide professional and constructive reviews. We are encouraged by the positive comments on the significance of our work in physical science and domain adaptation community. All suggestions will be carefully incorporated into final version.
>
> Q1. The name of 'physics-preserved'.
>
> We term our method as "physics-preserved" OT to emphasize its core mechanism and the physical constraints is not limited to $G_s$.
>
> (1)	As discussed in Sec. 4.2, we aim to characterize the target distribution and its underlying physical structure by a mapping. If the distributions match perfectly, standard OT can yield valid solutions. However, when distributions do not align completely (common scenario in practice), the pushforward distribution serves as an approximation of the target. The POTT is introduced to guide this approximation process in a physically meaningful way. This is why we name our method "physics-preserved" OT.
>
> (2)	The physical constraints in Eq. (10) are inherently problem-dependent. As our work focuses on developing a general method, we use $G_s$ as the basic constraint, which is universally accessible in most tasks. But in applications with additional prior knowledge (e.g., potential structures of the operator), more precise constraints can be designed.
>
> Q2. Relationship between proposed method and infinite-dimensional OT.
>
> In fact, the definitions and derivations of OT and its dual formulation are discussed in general separable metric spaces, which inherently includes infinite-dimensional function spaces. Thus, the definition of POTT is mathematically well-defined and not limited to finite-dimensional spaces. While the mentioned "OT on infinite-dimensional function space H" (denoted as OT-inf) explicitly instantiates the function space, it does not conflict with our definition.
>
> The key distinction lies in the practical implementation. In standard OT, the function variables (k, u) are directly discretized into tensors via sampling, leading to the empirical estimation form in Eq. (13). In contrast, the empirical estimation of OT-inf relies on properties of the specific function space H, such as the kernel trick. While OT-inf is more precise in definition, its computation requires restricting H to spaces like RKHS for tractability, which may limit the applicability of POTT.
>
> Q3. Explanation of learning procedure and computational cost.
>
> Algorithm: Transferring $G_s$ to $G_t$
>
> >1. Input: source data $D_s$, pretrained model $G_s$, target data $D_t$ , batchsize B
>
> >2. Initialize $T_\theta$, $f_\phi$
>
> >3. For $N_{11}$ steps:
>
> >4.    Update $\theta$ to minimize the first part in Eq. (13) for $N_{12}$ steps
>
> >5.    Update $\phi$ to maximize the first part in Eq. (13) with $\theta$
>
> >6. End
>
> >7. For $N_2$ steps:
>
> >8.    Update $\eta$ to minimize the second part in Eq. (13) with $\theta$ and $\phi$
>
> >9. End
>
> >10. Output: $G_\eta$ as approximation of $G_t$.
>
> Thus, the entire model transfer process of POTT requires O(($N_{11}$⋅$N_{12}$+$N_{2}$)B) operations. Typically, $N_{12}$ is set to 10. While OT-based methods are generally costly, they offer a viable strategic trade-off in scenarios where data acquisition is constrained or costly (e.g., weather forecasting, medical diagnosis) by leveraging algorithms and computation to reduce dependency on large-scale target-domain data.
>
> Q4. Applications for this problem setting.
>
> Thank you for highlighting this important point. This problem setting indeed has broad applications in various problem scenarios. In weather forecasting, for instance, differential equations are used to approximate the weather systems, and operator networks are trained on collected data to fit these systems. Generally, more data is available in developed regions  while less data is available in less developed regions. As geographical conditions vary across regions, directly deploying models across regions can reduce their performance. Thus, introducing transfer learning into cross-domain model deployment provides critical value, which highlights the importance of transfer learning in achieving technology and development equity around the world.
>
> Q5. Impact of domain shift and target samples on accuracy.
>
> Our experimental design systematically evaluates how domain shift and target sample amounts jointly influence model accuracy. Here’s a concise summary:
>
> (1)	Large or moderate domain shift reduces the effectiveness of finetuning (FT), while POTT significantly improves the accuracy. Minor domain shift allows FT to perform well, and POTT still offers improvements. Extremely small shift (overlapping subdomains) makes transfer unnecessary, while extremely large shift make model transfer infeasible.
>
> (2)	Abundant target samples make FT or even supervised training sufficient, with marginal gains from POTT. Limited target samples see POTT outperform FT largely. Very few target samples challenge all transfer methods.

---

### Decision · Program_Chairs · 2025-05-01

**Decision:**

Reject

**Comment:**

This paper presents a Physics-preserved Optimal Tensor Transport (POTT) that aims at addressing transfer learning problems for solving differential equations using operator learning approaches. The objective is to correct the operator bias while preserving physics knowledge to adapt to the target domain. Experiments on diverse PDE a proposed.

During the evaluation, reviewers have identified the following strengths:
- interesting extension of standard domain adaptation techniques that be used beyond physics applications
- the paper addresses the significant problem.
- reasonable approach
and the following weaknesses:
-method difficult to understand, an algorithm would help to better understand the procedures
-computational cost may be high, evaluation limited to synthetic datasets
-the method does not enforce PDE constraints.

During rebuttal, authors answered to the reviewers' comments allowing to provide clarifications on the approach.
After rebuttal, Reviewers  BWbB and NHBg updates their scores to weak accept.

During discussion, there is a general consensus for saying that the paper is interesting and sound, but there are still some small reservations on the presentation and the absence of experimental evaluation with physics-informed operator models enforcing PDE remains an important weakness.
The paper is borderline, after discussion and considering other papers, the contribution is evaluated as being below the acceptance bar.
Rejection is then proposed.